# The Role of Speckle Tracking Echocardiography in the Evaluation of Common Inherited Cardiomyopathies in Children and Adolescents: A Systematic Review

**DOI:** 10.3390/diagnostics11040635

**Published:** 2021-04-01

**Authors:** Dan M. Dorobantu, Curtis A. Wadey, Nurul H. Amir, A. Graham Stuart, Craig A. Williams, Guido E. Pieles

**Affiliations:** 1Children’s Health and Exercise Research Centre (CHERC), University of Exeter, Exeter EX1 1TX, UK; dd389@exeter.ac.uk (D.M.D.); cw694@exeter.ac.uk (C.A.W.); 2Population Health Sciences Department, University of Bristol, Bristol BS8 2PS, UK; nurul.amir@bristol.ac.uk (N.H.A.); Graham.Stuart@nhs.net (A.G.S.); Guido.Pieles@bristol.ac.uk (G.E.P.); 3Faculty of Sport Science and Recreation, Universiti Teknologi MARA, Perlis Branch, Arau Campus, Perlis 40450, Malaysia; 4National Institute for Health Research (NIHR) Cardiovascular Biomedical Research Centre, Congenital Heart Unit, Bristol Heart Institute, Bristol BS2 8ED, UK; 5Institute of Sport, Exercise and Health, University College London, London W1T 7HA, UK

**Keywords:** speckle tracking echocardiography, pediatric inherited cardiomyopathy, dilated cardiomyopathy, hypertrophic cardiomyopathy, left ventricular non-compaction, arrhythmogenic cardiomyopathy

## Abstract

Speckle tracking echocardiography (STE) has gained importance in the evaluation of adult inherited cardiomyopathies, but its utility in children is not well characterized. We conducted a systematic review to evaluate the role of STE in pediatric inherited cardiomyopathies. PubMed, EMBASE, Web of Science, Scopus, CENTRAL and CINAHL databases were searched up to May 2020, for terms related to inherited cardiomyopathies and STE. Included were dilated cardiomyopathy (DCM), hypertrophic cardiomyopathy (HCM), left ventricular non-compaction (LVNC) and arrhythmogenic cardiomyopathy (ACM). A total of 14 cohorts were identified, of which six were in DCM, four in HCM, three in LVNC and one in ACM. The most commonly reported STE measurements were left ventricular longitudinal strain (S_l_), circumferential strain (S_c_), radial strain (S_r_) and rotation/torsion/twist. S_l_, S_c_ and were abnormal in all DCM and LVNC cohorts, but not in all HCM. Apical rotation and twist/torsion were increased in HCM, and decreased in LVNC. Abnormal STE parameters were reported even in cohorts with normal non-STE systolic/diastolic measurements. STE in childhood cardiomyopathies can detect early changes which may not be associated with changes in cardiac function detectable by non-STE methods. Longitudinal and circumferential strain should be introduced in the cardiomyopathy echocardiography protocol, reflecting current practice in adults.

## 1. Introduction

The publication of recent registry-based reports has led to an increasing interest in the pediatric inherited cardiomyopathies, including hypertrophic cardiomyopathy (HCM), dilated cardiomyopathy (DCM), left ventricle non-compaction cardiomyopathy (LVNC) and arrhythmogenic cardiomyopathy (ACM) [1,2,3]. Although rare [1,2,3], they carry a significant mortality and morbidity burden: DCM is the main indication for pediatric heart transplant [1]; 7% of older children or adolescents with HCM died or received a heart transplant [2]; isolated LVNC, diagnosed in late childhood or adolescence, has 6% mortality at 5 years [3]; pediatric ACM has been associated with a high prevalence of life threatening arrhythmia at first presentation, compared to adults [4].

Diagnosis and follow-up of cardiomyopathy patients can be complicated by heterogeneity in clinical presentation and etiology, as well as overlap with acquired causes, such as myocarditis, or normal variants, such as athletes. Two dimensional speckle tracking echocardiography (STE) is used routinely in the management of adult cardiomyopathies and has been associated with clinical outcomes in HCM [5], DCM [6], LVNC [7] and ACM [8], aiding screening and early diagnosis [9,10,11,12,13]. Despite overwhelming data from the adult cardiomyopathy practice, STE has only recently started to gain importance on the pediatric side. 

The aim of this study is to conduct a systematic review and meta-analysis of studies reporting STE measurements in pediatric inherited cardiomyopathies. The objectives of the review included understanding this technique’s current use in clinical practice, as compared to conventional echocardiographic techniques, and how its role in this field could be expanded in the future.

## 2. Materials and Methods

### 2.1. Study Design and Aims

This study was designed according to the Preferred Reporting Items for Systematic Reviews and Meta-Analyses (PRISMA) guidelines [14]. The protocol was submitted to PROSPERO (CRD42020170433) during abstract screening. The primary aim of the study was to evaluate the role of STE in the diagnosis and follow-up of common inherited cardiomyopathies presenting in later childhood, beyond infancy. The secondary aim was to compare STE with conventional techniques in the same setting. 

### 2.2. Study Eligibility

Inclusion criteria were: (1) Children or adolescents (mean or median age < 18 years); (2) Condition of interest: HCM, DCM, LVNC, ACM; (3) Echocardiography at rest with any STE measurement; (4) full text paper published in a peer-reviewed journal.

Exclusion criteria were: (1) Mean/median age under 2 years (or predominantly “neonate”, “infant” or “toddler”); (2) mean/median age over 20 years; (3) other cardiomyopathies (e.g., muscular dystrophies, mitochondrial diseases, amyloidosis, degenerative, metabolic or infectious cardiomyopathies, oncological treatment related disease, congenital heart disease, myocarditis, arrhythmic); (4) critically ill patients (New York Heart Association/Ross functional class IV, on inotrope support, on ventricular assist devices or circulatory assist); (5) MRI derived strain imaging; (6) Tissue Doppler strain; (7) 3D/4D STE studies, with no 2D STE data; (8) conference abstracts without full text paper available. 

For studies with a mean/median reported age between 2–6 years or 18–20 years, those where the reported age distribution corresponded to mostly infants (<1 year)/toddlers (<3 years) or adults, respectively (>18 years), have been excluded, through consensus of the team. If mixed cardiomyopathies are reported, the study was excluded if more than 10% of the study population had another diagnosis than the four stated in the inclusion criteria. There were no language restrictions. 

### 2.3. Database Search

PubMed, EMBASE, Web of Science, Scopus, Cochrane CENTRAL and EBSCO CINAHL databases were searched from inception to May 2020. The search strategy included the combined terms relating to the common inherited cardiomyopathies (i.e., HCM, DCM, LVNC, ACM or “cardiomyopathy”), pediatric population, echocardiography and speckle tracking imaging. The full search strategy, pre-published on PROSPERO, is available in Appendix A. 

### 2.4. Study Selection and Data Extraction

Title and abstract and full text screening were performed using the Covidence^®^ platform (Veritas Health Innovation Ltd, Melbourne, Australia), independently, by two reviewers (D.M.D. and C.W.). Data extraction was conducted independently by D.M.D. and C.W. which included study and methodology description, patient/control group demographics, STE and conventional echocardiography measurements, which were checked for discrepancies. In all of the steps above, a third reviewer (G.P.) arbitrated disagreements, and any discrepancies were resolved through team consensus. Corresponding authors were contacted and asked for any non-reported outcome data for their respective cohorts, relevant to this analysis.

### 2.5. Risk of Bias Assessment

Risk of bias was assessed independently by D.M.D. and C.W. using the Newcastle-Ottawa scale for case control studies [15]. Risk of bias assessment was done for 13 papers, since two studies [16,17] presented the same cohort. Studies were also graded into “low”, “medium” and “high” risk of bias, taking into consideration the Newcastle-Ottawa scales, through team consensus. More detail on the risk of bias assessment can be found in Appendix A. 

Publication bias was assessed visually by funnel plots of standardized mean difference (SMD) and standard error of SMD, and using the Egger test, acknowledging that this method could result in over-estimation of risk of publication bias when SMD is used [18].

### 2.6. STE and Conventional Echocardiographic Measurements 

For the purpose of plotting and analysis, STE measurements were grouped as follows: longitudinal strain—S_l_, including apical four chamber (A4C) and global longitudinal strain (GLS); circumferential strain—S_c_, including basal, mid and apical circumferential strain; radial strain—S_r_ including basal, mid and apical radial strain; twist or torsion; basal rotation; and apical rotation. The main non STE measurements were left ventricular ejection fraction (LVEF), septal S’, E/A and E/E’ ratios. Details on all echocardiographic data extracted are found in Appendix A. 

### 2.7. Statistical Analysis

Frequencies are given as numbers and percentages, continuous values as means and standard deviation (SD). Numerical values extraction, transformation and calculation of missing statistical parameters are detailed in Appendix A. 

Data were summarized according to the Synthesis without meta-analysis (SWiM) guidelines [19], and presented by cardiomyopathy type. Given the high heterogeneity of the results, continuous variables were reported as range, or as individual data points, not as means. Albatross plots were used to compare and summarize the effect size, sample size and *p* values, due to the applicability in diversely reported studies [20]. Actual SMD values are reported in tabular form (Appendix A), with SMD contours in the Albatross plots assuming all studies have a 1:1 disease to control ratio, exclusively for in-between study comparisons’ visual representation. Statistical analyses were conducted using STATA/SE 12 (StataCorp LP, College Station, TX, USA).

## 3. Results

After screening 949 papers, with 102 full text articles assessed for eligibility, a total of 14 papers (14 cohorts) fulfilled the inclusion criteria and had no exclusion criteria (Figure 1). One paper included both a DCM and a HCM cohort, and was considered as two cohorts [21], while two papers reported different measurements in the same cohort at two different times [16,17]. Thirteen were case control studies with healthy controls, one had HCM with and without high risk genotype as participants/controls, and only one had partial follow-up after baseline evaluation.

### 3.1. Selected Studies 

Characteristics of included studies and reported STE measurements are summarized in Table 1. Of the 14 cohorts, six were in DCM, four in HCM, three in LVNC and 1 in ACM. The number of participants ranged from 10–50, with a mean of 28 (10). Mean age ranged from 4.5–15 years, and males accounted from 46–86% of samples. 

Most commonly reported non-STE cardiac systolic and diastolic function measurements were LVEF (*n* = 14), septal S’ (*n* = 5), E/A ratio (*n* = 7) and E’/E’ ratio (*n* = 9), summarized in Appendix A. 

### 3.2. Dilated Cardiomyopathy

Mean age in six DCM studies ranged from 4.5–10 years, male percentage from 48% to 70%. Figure 2A shows a summary of STE findings in these studies: there were less negative S_l_ (−11% to −15% in 3 studies), S_r_ (−15.7% and −18.5% in 2 studies) and S_c_ (−9.5% in one study); one study reported reduced twist (0.3° vs. 10.9°), mostly due to reduced apical rotation (0.9° vs. 5.9°). An additional study (not shown in Figure 2A) reported only mechanical dyssynchrony measurements derived from STE, of which we mention maximum S_l_ and S_r_ delays, both significantly higher in DCM. Mean differences, SMDs and *p* values are detailed in Appendix A. All DCM cohorts reported reduced LVEF (mean ranging from 26.5% to 49.7%), with three reporting reduced septal S’, two reporting increased E/E’ ratio and thee reporting comparable E/A ratio (Figure 3A). 

### 3.3. Hypertrophic Cardiomyopathy

Mean age in four HCM studies ranged from 6.1–14.1 years, male percentage from 60% to 85%. Figure 2B shows a summary of STE reported findings in the HCM cohorts: S_l_ was normal in one study (−21.3%) and less negative in two (−15.8% and −16.7%); S_c_ was abnormal in one study (−17.5%) and normal in one study (−22.2%); two studies reported rotational mechanics, showing increased twist, with more positive apical rotation (11.7° vs. 5.3° and 13.9° vs. 8.8°) and torsion, with more negative basal rotation (−8.7° vs. −4.9° and 2.8°/mm vs. 1.9°/mm), respectively. Mean differences, SMDs and *p* values are detailed in Appendix A. All studies included patients with normal or supra-normal LVEF (mean ranging from 64–67.4%), one study reported normal septal S’ values, E/E’ was increased in one study and normal in one study, while the E/A ratio was reported normal in two studies (Figure 3B).

### 3.4. Left Ventricular Non-Compaction

Mean age in four LVNC studies ranged from 7.2–12.1 years, male percentage from 50% to 80%. S_l_ ranged from −15.3% to 18% in four studies. All studies reported less negative S_c_ (−18.1% to −19.1% for basal, −16.3% to −24.6% for mid and −14.9% to −20.6% for apical). S_r_ was also reduced in two studies, for all three segments. Two studies reported rotational mechanics, showing reduces twist (10° vs. 15.1° and 3.8° vs. 13.5° respectively), both due to reduced apical rotation. Mean differences, SMDs and *p* values are detailed in Appendix A. All studies reported normal or mildly reduced LVEF (mean ranging from 54% to 68.9%), one study reported normal septal S’ values, three studies reported normal or mildly increased E/E’ values and one study reported normal E/A ratio (Figure 3C).

### 3.5. Arrhythmogenic Cardiomyopathy

Only one study reported STE measurements in children with ACM, with a mean age of 15 years, 68% male. The main findings were a less negative RV S_l_, both global (−21% vs. −25%) and free wall (−19% vs. −24%), more markedly in the apical segments, despite similar FAC, TAPSE and RV S’ values (Appendix A). 

### 3.6. Risk of Bias Assessment

#### 3.6.1. Publication Bias

Overall risk of publication bias for any measurement was low (Egger *p* value 0.08), as presented in Figure 4. Risk of publication bias analysis was also repeated in subgroups by cardiomyopathy and STE measurement types, and found no high risk of bias (all Egger *p* values > 0.05).

#### 3.6.2. Study Methodology Bias

Most studies, 9/13, were graded as medium risk of bias, 3/13 as low risk and 1/13 as high risk (Figure 5). The three studies with low risk either managed to perform a blinded evaluation of STE images, clearly report a comparable proportion of patients/controls with unusable STE images, or both. One study offers limited to no information on selection of cases and controls, and scored no point in the “Selection” item of the scale. Studies obtaining lower scores in the Selection item either failed to detail how the patients were selected (i.e., consecutive or not) or, in one instance, did not use healthy controls. These findings do not reflect poor study methodology or quality, but rather inherent limitations of echocardiographic case-control studies, where consecutive case selection, community controls and blinding of evaluation are not always feasible. 

## 4. Discussion

This systematic review on the use of 2D STE in the main forms of pediatric inherited cardiomyopathy, DCM, HCM, LVNC and ACM, revealed several important findings. Firstly, only 14 studies were found (a single study in ACM), with high heterogeneity by cardiomyopathy type and STE measurement. This observation is despite accumulating evidence of the usefulness of STE in adult cardiomyopathy, both in early diagnosis [9,10,11,38] and outcome prediction [5,6,7,8,39]. Secondly, changes in STE parameters were described in pediatric cohorts where systolic or diastolic ventricular function were comparable to the healthy controls, demonstrating there is added value of STE over conventional echocardiography in the evaluation of children with cardiomyopathies. 

When interpreted in the context of existing adult inherited cardiomyopathy data, the findings of this systematic review support the use of measurement of STE systolic function parameters in the evaluation of children. When expertise is available, rotational mechanics parameters, in addition to conventional echocardiographic techniques, could offer further details, especially in HCM and LVNC. 

### 4.1. Pediatric Dilated Cardiomyopathy

All DCM studies reported reduced LVEF and abnormal LV strain values (longitudinal, circumferential and radial), when compared to controls. Adult data supports the notion that abnormal LV strain is predictive of cardiac events [6], and STE has been shown to be a more sensitive tool in assessing changes in LV systolic function than LVEF alone in children with other causes of LV dysfunction, such as anthracycline toxicity [40]. Thus, it is reasonable to include measurements of LV strain in the diagnosis and follow-up of pediatric DCM, in addition to conventional parameters. Two studies showed abnormal rotation [22] and dys-synchrony parameters [24], but their clinical significance has not yet been established. 

### 4.2. Pediatric Hypertrophic Cardiomyopathy

All HCM studies reported normal/supranormal LVEF, with one reporting abnormal E/E’. Just two studies reported LV deformation parameters: normal S_c_ (less negative in apical segments) in one [28] and abnormal S_l_ in the second [21]. Both studies reporting rotational mechanics showed that basal rotation was less pronounced, while apical rotation was increased compared to controls, also observed in adults with HCM [41], but not the apical form, where apical rotation is reduced [42]. Rotational mechanics are thus a potential tool for HCM screening in children, and a promising avenue for research. Both Pieles et al. [29] and Forsey et al. [28] linked STE parameters to genotype variants, proposing a role of STE in screening and risk stratification. As opposed to adult HCM, where LV systolic deformation is reduced more frequently, [5] in children this was not seen consistently.

### 4.3. Left Ventricular Non-Compaction Cardiomyopathy

LV longitudinal, circumferential and radial strain were found to be abnormal compared to controls, as well as apical rotation, and subsequently twist and torsion. Basal rotation was normal, as opposed to adults, where decreased basal rotation was reported, in addition to rigid body rotation [7]. These findings support the notion that changes in myocardial mechanics are seen as early as childhood, and progress through adulthood, making STE a very useful tool for follow-up and screening. 

### 4.4. Arrhythmogenic Cardiomyopathy

ACM was only reported in one cohort, despite growing evidence that early pediatric cases pose significant issues in early diagnosis, follow-up and risk stratification [4,43]. Recent adult data from the Dutch ACM Registry also showed that myocardial mechanics are altered in gene carriers not meeting diagnosis criteria, as well as an early potential sign of disease progression [12,13]. Nevertheless, given the importance of early ACM diagnosis to prevent sudden cardiac death, more research is needed to determine whether STE can accurately identify children with ACM.

### 4.5. Current Issues and Future Directions

We found few studies reported similar STE measurements in the same inherited cardiomyopathy type (three at most or any parameter in the same disease), highlighting the urgent need for a standardized protocol for children. This is further emphasized by the overall high risk of bias observed (only 3/14 of studies achieved a low risk), which may not reflect poor study quality, but rather inherent limitations in reporting, compounded by heterogeneous center practices. Nevertheless, most STE parameters were found to be altered in inherited cardiomyopathies when compared to healthy controls, even when reported as secondary findings. 

From a clinical point of view, the data summarized by this systematic review suggest that, while it is reasonable to report longitudinal strain alongside conventional systolic function parameters, there is a stringent need for prospective, longitudinal studies in pediatric ICCs, to determine the exact benefits of adding STE to the standard of care. In addition, more data are needed to establish the role of rotational mechanics and dyssynchrony parameters in clinical practice.

### 4.6. Limitations

The main limitation comes from the small number of studies included, with very different populations and measurements, limiting the report to a synthesis without meta-analysis. All included studies had inherent patient selection biases, and this extends to our analysis. In addition, studies with negative results tend not to be reported, and this can cause publishing bias. Inter-vendor variability of STE measurements between studies was expected [44], and for this reason standardized mean differences were used in comparisons, instead of reported values. The analysis was also limited in its scope, with at most three cohorts reporting the same measurement in the same condition. Some items in the Newcastle-Ottawa scale for case-control studies were difficult to address in most imaging studies, due to inherent methodological flaws (e.g., impossibility of true blinding, tendency to select controls from healthy hospital patients), and, as such, the risk of bias could have been over-estimated. To balance this, we also used a semi-qualitative grading of studies (low/medium/high), taking scale results into consideration. Studies reporting cardiomyopathies secondary to other pathology have not been included, to try to minimize diagnostic heterogeneity. However, we acknowledge that in the studies included, some diagnostic misclassification may have occurred.

## 5. Conclusions

There is limited data on STE use in pediatric inherited cardiomyopathy, especially in ACM, with significant variety in reported STE measurements. The available data indicates that there are early changes during childhood measurable by STE, with or without abnormal ventricular systolic or diastolic function by non-STE methods. Longitudinal and circumferential strain should be introduced in the cardiomyopathy echocardiography protocol, mirroring adult practice. Myocardial mechanics, especially rotational parameters in LVNC and HCM show promise as early screening tools, and should be used when the expertise is available.

## Figures and Tables

**Figure 1 diagnostics-11-00635-f001:**
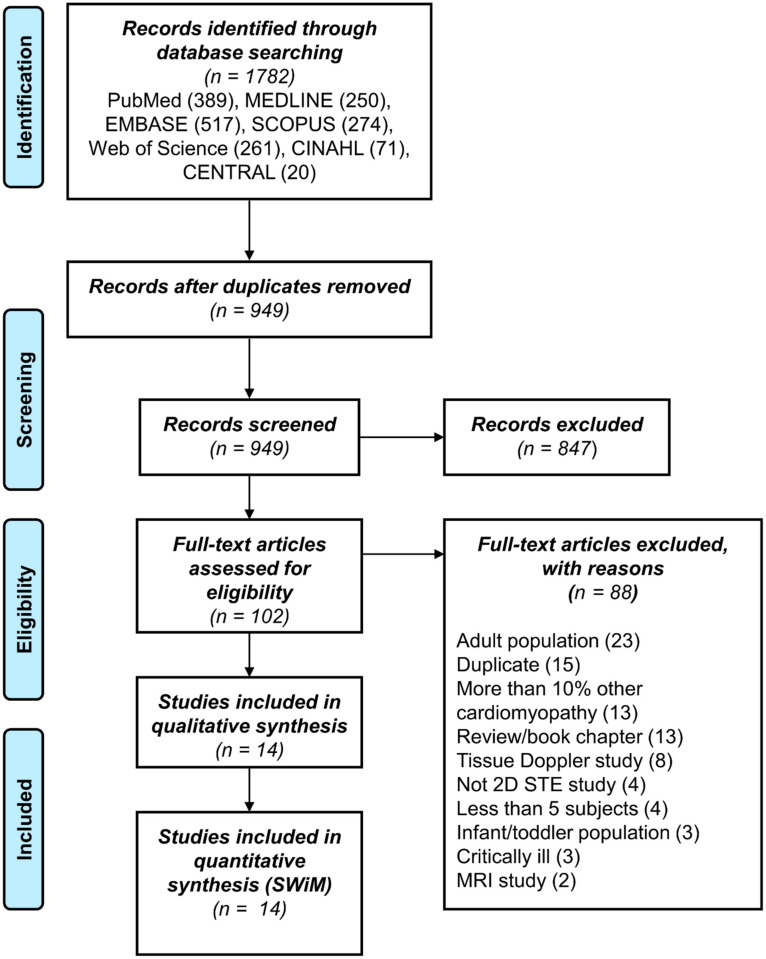
PRISMA flowchart of screened, included and excluded studies.

**Figure 2 diagnostics-11-00635-f002:**
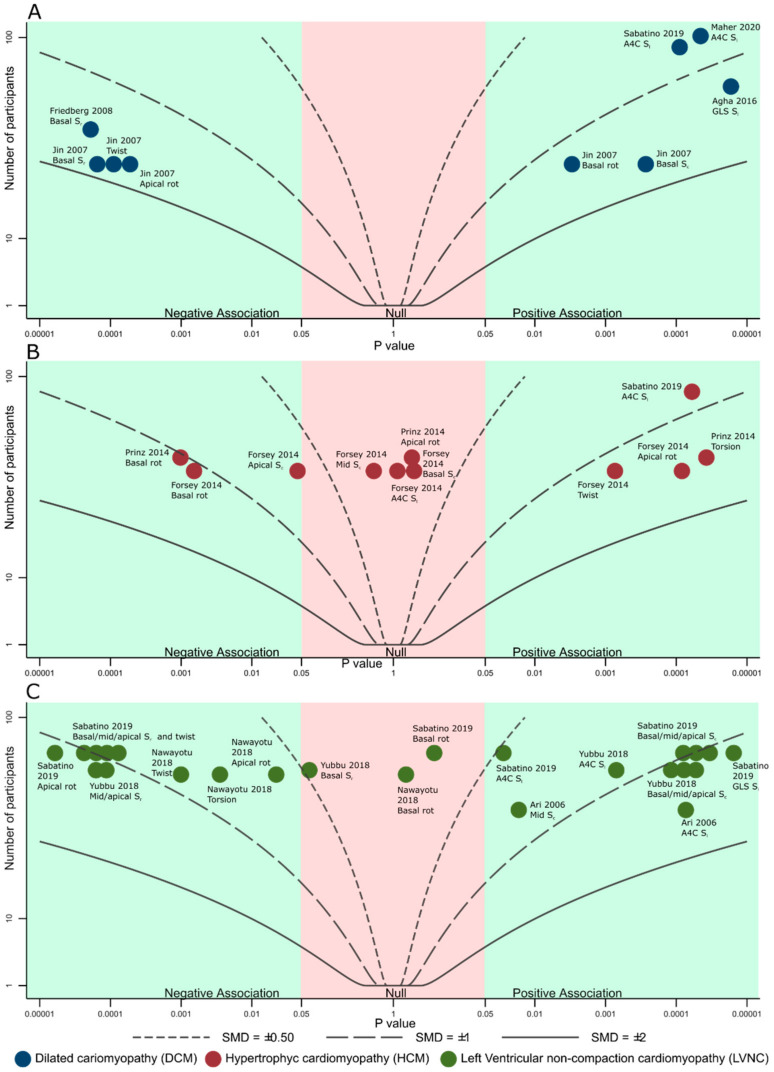
Albatross plot showing estimated standardized mean differences contours (curved lines), *p* values for difference between inherited cardiomyopathy and controls (horizontal axis) and study sample size (vertical axis) of speckle tracking measurements by cardiomyopathy type in (**A**). Dilated cardiomyopathy; (**B**). Hypertrophic cardiomyopathy (**C**). Left ventricular non-compaction. Negative association–lower values (including more negative) associated with disease; positive association–higher values (including less negative) associated with disease. A4C, apical four chamber view; GLS, global longitudinal strain; rot, rotation; S_l_, longitudinal strain; S_c_, circumferential strain; S_r_, radial strain.

**Figure 3 diagnostics-11-00635-f003:**
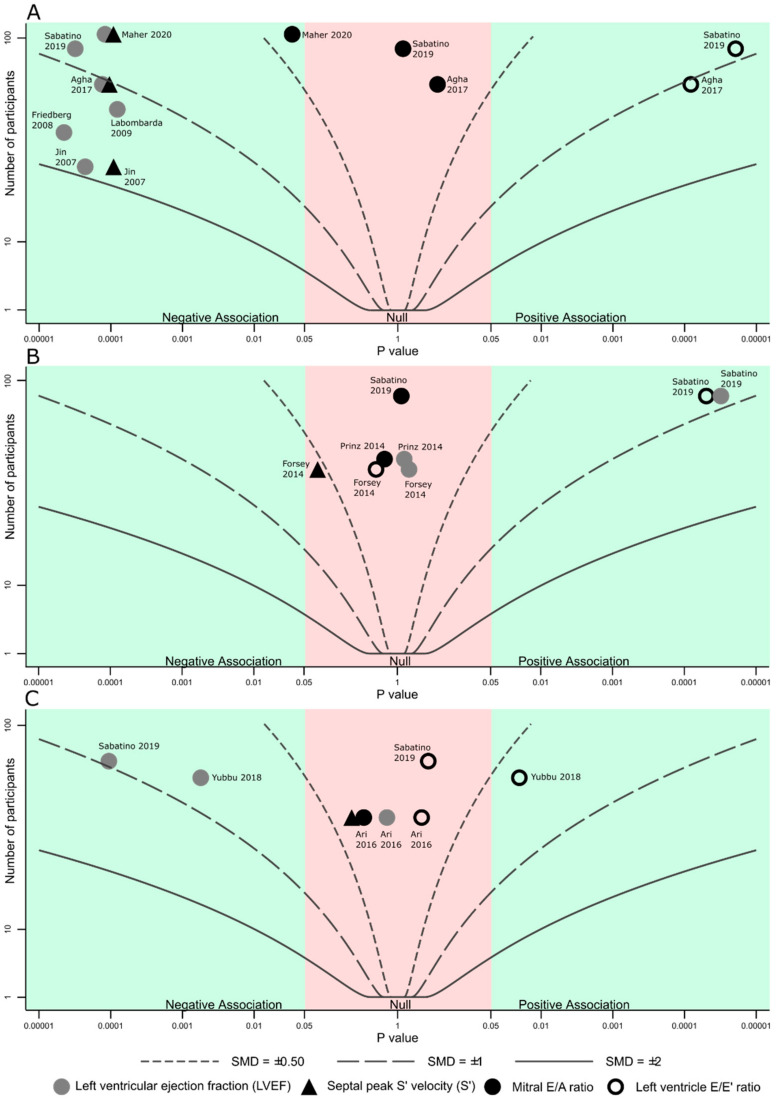
Albatross plot showing estimated standardized mean differences contours (curved lines), *p* values for difference between inherited cardiomyopathies and controls (horizontal axis) and study sample size (vertical axis) of conventional echocardiographic measurements by cardiomyopathy type in (**A**). Dilated cardiomyopathy; (**B**). Hypertrophic cardiomyopathy (**C**). Left ventricular non-compaction. Negative association–lower values (including more negative) associated with disease; positive association–higher values (including less negative) associated with disease.

**Figure 4 diagnostics-11-00635-f004:**
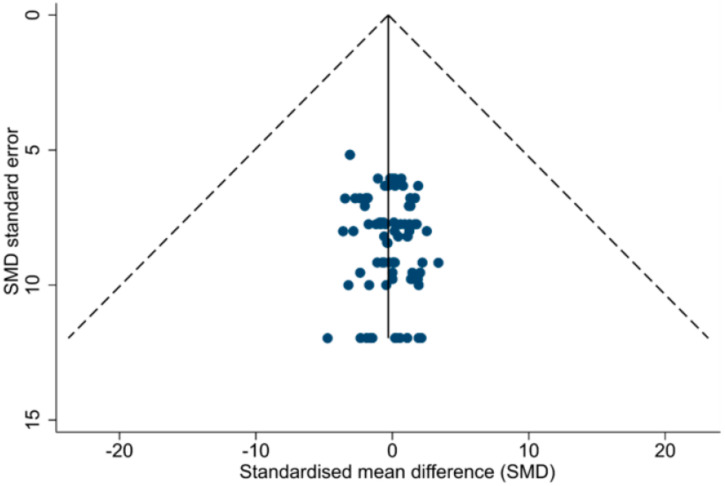
Funnel plot of Standardized Mean Difference (SMD) and SMD standard error with pseudo 95% CI limits for all study measurements, showing no significant asymmetry, which suggests low risk of publication bias. Eggert test *p* value is 0.08.

**Figure 5 diagnostics-11-00635-f005:**
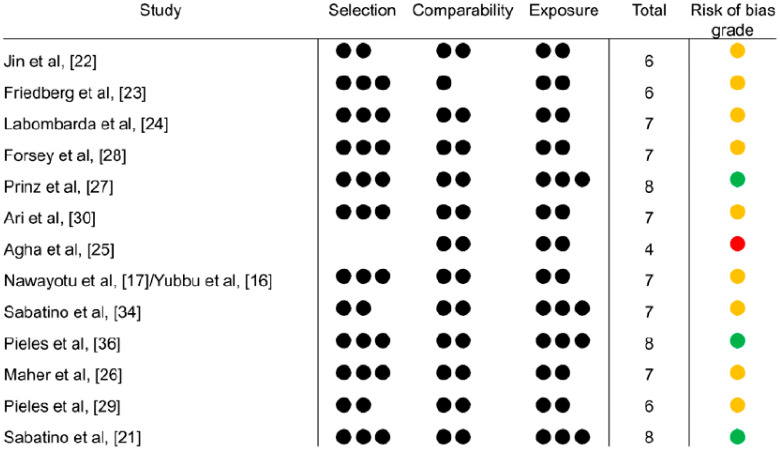
Newcastle Ottawa scale risk of bias evaluation results. Total maximum is 9 points, Selection is 3 points, Comparability is 2 points and Exposure is 4 points. Semi qualitative subjective risk of bias grading is green for low risk, orange for some concerns and red for high risk.

**Table 1 diagnostics-11-00635-t001:** Summary of included studies.

Source	Inherited Cardiomyopathy Diagnosis Criteria	Study Size, No.		Mean Age, y (sd)		Male, No. (%)		STE Measurements
		Cardiomyopathy	Controls	Cardiomyopathy	Controls	Cardiomyopathy	Controls	
Dilated cardiomyopathy (DCM)								
Jin et al. [22]	Not specified	10	17	5.9 (4.6)	6 (4.2)	7 (70)	12 (70.5)	basal S_c_, basal S_r_, rotational mechanics
Friedberg et al. [23]	LVEDD > 2 SD LVEF < 55%	24	16	10 (6)	10 (6)	11 (45.8)	4 (25)	mid S_r_
Labombarda et al. [24]	LVEDD > 2 SD LVEF < 50%	25	25	7.8 (4.9)	7.8 (4.9)	17 (68)	17 (68)	dyssynchrony parameters
Agha et al. [25]	Not specified	32	32	5.1 (4.4)	5.88 (3.92)	19 (59.4)	20 (62.5)	LV GLS
Sabatino et al. [21]	LVEDD > 2 SD LVEF < 50%	44	45	5 (5.2)	10.4 (4.7)	21 (47.7)	18 (40)	LV A4C S_l_
Maher et al. [26]	Not specified	50	50	4.5 (1.8)	4.6 (2)	29 (58)	27 (54)	LV A4C S_l_
Hypertrophic cardiomyopathy (HCM)								
Prinz et al. [27]	Clinical, ECG and echocardiographic criteria for hypertrophy, without other causes	24	20	14.1 (5.5)	14.1 (5.5)	14 (58.3)	14 (70)	rotational mechanics
Forsey et al. [28]	Documented family history and genotype, with no hypertrophy	14	28	9.8 (4.5)	9.8 (4.4)	12 (85.7)	23 (82.1)	LV A4C S_l_, basal/mid/apical S_c_, rotational mechanics
Sabatino et al. [21]	Wall thickness in any segment > 2.5 Z score and normal/increased LVEF	40	45	10.9 (5.5)	10.4 (4.7)	24 (60)	18 (40)	LV A4C S_l_
Pieles et al. [29]	Clinical, ECG and echocardiographic criteria, without other causes	25	0	6.1 (4.5)		19 (76)		LV A4C S_l_, basal S_c_, basal S_r_
Left ventricular non-compaction (LVNC)								
Ari et al. [30]	Jenni [31] criteria	20	20	12.1 (3.3)	11.8 (3.1)	16 (80)	16 (80)	LV A4C S_l_, mid S_c_
Yubbu et al. [16] (Nawayotu et al. [17] first study, same cohort, reports 28 pts)	Jenni [31] Stollberger [32] and Chin [33] criteria	30	30	7.2 (5.6)	8.4 (5.3)	15 (50)	14 (46.7)	LV A4C S_l_, basal/mid/apical S_c_, basal/mid/apical S_r_ in Yubbu et al. [16] rotational mechanics in Nawayotu et al. [17]
Sabatino et al. [34]	Peterson [35] CMR criteria	23	47	11.3 (5.3)	11.1 (5.3)	12 (60)	24 (40)	LV A4C S_l_, LV GLS, basal/mid/apical S_c_, basal/mid/apical S_r_, rotational mechanics
Arrhythmogenic cardiomyopathy (ACM)								
Pieles et al. [36]	ACM modified Task Force criteria [37]	38	35	15 (3)	13 (4)	26 (68.4)	29 (82.9)	RV S_l_

A4C, apical 4 chamber view; CMR, cardiac magnetic resonance; GLS, global longitudinal strain; LV, left ventricle; LVEDD, left ventricular end-diastolic diameter; LVEF, left ventricular ejection fraction; RV, right ventricle; S_c_, circumferential strain; S_l_, longitudinal strain; S_r_, radial strain; STE, speckle tracking echocardiography.

## Data Availability

Data available on request.

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
