# Peer review of "The Role of Speckle Tracking Echocardiography in the Evaluation of Common Inherited Cardiomyopathies in Children and Adolescents: A Systematic Review"

_diagnostics, 2021, doi:10.3390/diagnostics11040635_

Round 1

Reviewer 1 Report

This is a well written paper, regarding an important question in the prediction of disease and possibly risk in pediatric patients with genetic cardiomyopathies. 

I feel the research was done appropriately, the drawback of this analysis is the lack of consistency (and uniform protocol) in the strain measures used in the different papers include in this analysis. Therefore the conclusions of your paper are inherently debatable, this is appropriately reflected in the discussion in my opinion.

In the introduction I feel two references should be added

1) Taha et. al JACC Cardiovasc Imaging 2020 Nov 13;S1936-878X(20)30918-9. Early Mechanical Alterations in Phospholamban Mutation Carriers: Identifying Subclinical Disease Before Onset of Symptoms. (A form of ARVC)

2) Taha et al. JACC Cardiovasc Imaging 2020 Feb;13(2 Pt 2):631-634. Evaluation of Disease Progression in Arrhythmogenic Cardiomyopathy: The Change of Echocardiographic Deformation Characteristics Over Time.

Author Response

We agree that there are inherent limitations in a meta-analysis of this kind, due to variability in protocols, but for this reason we have used the standardised mean differences rather than reported values, and this allows for a more uniform analysis of the data. The conclusions can of course be discussed and interpreted by the readers, but are nevertheless supported by good quality data, derived from appropiate analyses.

Additionally, we have included these two topical references in the Introduction (line 49), and added a brief discussion on them in the topic of our findings (lines 273-276)

Reviewer 2 Report

This is an interesting and well written review article sistematically analyzing the role of speckle tracking echocardiography in the evaluation of common inherited cardiomyopathies in children and adolescents. 

I have only a minor concern based on the reproducibility of left ventricular strain. I would like to invite the authors to include and discuss the article from Konstantinos et al (J Am Soc Echocardiogr 2015;28:1171-81)

Author Response

The issue of inter-vendor variability was considered in the analysis, by using SMD instead of raw reported values in all comparisons. This is now clear in the Limitations section (lines 299-301)